# Users' experiences of a pragmatic diabetes prevention intervention implemented in primary care: qualitative study

Navneet Aujla,[1] Thomas Yates,[2] Helen Dallosso,[3,4] Joe Kai[1]

¹Division of Primary Care, University of Nottingham, Nottingham, UK
²Diabetes Reseach Centre, College of Life Sciences, University of Leicester, Leicester, UK
³University Hospitals of Leicester, Leicester, UK
⁴NIHR Leicester Biomedical Research Centre, University of Leicester, Leicester, UK

**Correspondence to**
Professor Joe Kai;
joe.kai@nottingham.ac.uk

## ABSTRACT

**Objectives** To explore service-user and provider experience of the acceptability and value of the *Let's Prevent Diabetes* programme, a pragmatic 6-hour behavioural intervention using structured group education, introduced into primary care practice.

**Design** Qualitative interview-based study with thematic analysis.

**Setting** Primary care and community.

**Participants** Purposeful sample of 32 participants, including 22 people at high risk of diabetes who either attended, defaulted from or declined the intervention; and 10 stakeholder professionals involved in implementation.

**Results** Participants had low prior awareness of their elevated risk and were often surprised to be offered intervention. Attenders were commonly older, white, retired and motivated to promote their health; who found their session helpful, particularly for social interaction, raising dietary awareness, and convenience of community location. However attenders highlighted lack of depth, repetition within and length of session, difficulty meeting culturally diverse needs and no follow-up as negative features. Those who defaulted from, or who declined the intervention were notably apprehensive, uncertain or unconvinced about whether they were at risk of diabetes; sought more specific information about the intervention, and were deterred by its group nature and day-long duration, with competing work or family commitments. Local providers recognised inadequate communication of diabetes risk to patients. They highlighted significant challenges for implementation, including resource constraints, and facilitation at individual general practice or locality level.

**Conclusions** This pragmatic diabetes prevention intervention was acceptable in practice, particularly for older, white, retired and health-motivated people. However, pre-intervention information and communication of diabetes risk should be improved to increase engagement and reduce potential fear or uncertainty, with closer integration of services, and more appropriate care pathways, to facilitate uptake and follow-up. Further development of this, or other interventions, is needed to enable wider, and more socially diverse, engagement of people at risk. Balancing a locality and individual practice approach, and how this is resourced are considerations for long-term sustainability.

## Strengths and limitations of this study

► This study provides insights into how early implementation of a pragmatic diabetes prevention programme was experienced in practice, suggesting how engagement and the intervention might be developed or enhanced.

► Data were generated and analysed by researchers independent from those who developed or implemented the intervention.

► The purposeful sample reflected the demographic of the population who received the intervention, but also included respondents who defaulted from or declined the intervention, and who were educationally diverse; however, ethnic diversity was limited.

► Quantitative evaluation data for the whole intervention population are also needed, and will be reported separately.

## INTRODUCTION

Diabetes mellitus is one of the greatest healthcare challenges of this age, affecting millions of individuals globally.[1] Prevention, or delaying the onset of type 2 diabetes (T2DM), is thus a key public and healthcare priority.[2] Intensive lifestyle interventions are the cornerstone of diabetes prevention programmes[3] and address non-diabetic hyperglycaemia as one of the major risk factors for T2DM. In the UK, the NHS Diabetes Prevention Programme was recently launched as a national programme, with phased implementation planned across England for up to 100 000 at high risk of T2DM by 2020.[4] Early data from national pilot sites (2016–2017) have been promising, suggesting higher than expected uptake.[5]

The *Let's Prevent Diabetes* programme was developed with complex interventions guidance.[6 7] It is a 6-hour pragmatic behavioural intervention, adopting a structured education approach, delivered by two trained educators in a group format, using presentation of information and facilitation of group sharing

of experiences and interaction (further details reported elsewhere).[8–11] It uses a theory-driven, person-centred written curriculum, designed to target perceptions and knowledge of diabetes and diabetes risk as well as the motivational and volitional determinants of behaviour change, including self-efficacy, goal setting and self-monitoring. The programme is aimed at supporting individuals with a high risk of T2DM to better understand what their risk means and to set goals for weight loss, a healthy diet and increased physical activity. The programme was designed for a primary care setting and was evaluated in a randomised controlled trial.[8 10] This suggested that delivering this programme in primary care could lead to modest reductions in the risk of T2DM.[8] However, 23% of those consenting and randomised to the intervention did not attend the first 6-hour education session, and 71% failed to attend annual 'refresher' sessions.[10] Those 29% who completed two annual 3-hour refresher sessions reduced their risk of developing T2DM by up to 90%.[10] Similar poor uptake and adherence is reported in other community-based diabetes prevention programmes.[11–13]

The aim of diabetes prevention is to be effective, acceptable and feasible across the breadth of the population at high risk, rather than just in those willing to sustain attendance in controlled trial settings. Greater understanding of people's views of intervention delivered in the 'real world' is needed. Working with general practices in one English region, the *Let's Prevent Diabetes* programme was introduced into routine healthcare for a 1-year pilot period. In this qualitative study, we explored what helped or hindered engagement with this intervention; and its perceived value and acceptability in practice from the perspectives of those receiving or implementing the intervention.

## METHODS

### Context: implementation of *Let's Prevent Diabetes* programme

A federation of nine general practices in an East Midlands locality with a diverse population of 166 100 (40% aged 30–59 years) implemented the *Let's Prevent* intervention for 12 months. Educational resources (such as food models) were tailored to the local population, including Eastern European and Bengali communities, informed by presentation and discussion at community engagement events and consultation with patient and public involvement representatives. However, programme delivery was in English without translation of materials due to funding restrictions. Intervention sessions tailored to different ethnic groups were scheduled, but cancelled due to difficulty filling them, meaning all sessions were delivered to 'all comers'. A patient referral pathway was agreed after stakeholder meetings with general practitioners (GPs), practice nurses and practice managers. Six people with health or exercise facilitation backgrounds (eg, practice nurses, health promotion educator) were recruited and trained to deliver *Let's Prevent*, using a 2-day standardised and accredited programme for trainers at

Leicester Diabetes Centre.[14] The intervention was delivered to groups of 6–10 people, in a 6-hour session, in community settings, such as public library or community centre adjacent to a large primary care centre. A locality administrator coordinated delivery for 12 months. This included running database searches in the general practices to identify patients at elevated diabetes risk (routine glycated haemoglobin of 42–46.4 mmol/L (6.0%–6.4%) within the previous 24 months), sending letters from practices informing them of their elevated risk, with invitation to the *Let's Prevent Programme*. Where possible, this was also followed-up by a telephone call from the administrator to invitees.

### Sampling and data generation

We conducted one-to-one semi-structured interviews with people at high risk of T2DM invited to the *Let's Prevent* programme. A purposeful sample was sought to include a range of demographic, practice and those who attended, accepted invitation but did not attend, or declined the intervention. Interviews were facilitated with use of a topic guide, and explored views and experiences of the *Let's Prevent* intervention; barriers and facilitators to participating; and impact of attending the education session. This included seeking views about the process of being invited; motivation for attendance or reasons for non-attendance; the intervention's approach and relevance and reported behaviour change following participation. In parallel, we conducted semi-structured interviews and a focus group discussion with local stakeholders involved in the commissioning, facilitation and implementation of the intervention. We sought insights into barriers, enablers and contextual issues for the intervention, and explored views on its feasibility, acceptability, sustainability and effectiveness. A purposive sample was sought to reflect diversity of stakeholder experience and involvement in the intervention, including primary care clinical commissioning group (CCG) leads, intervention administrator, GPs, practice nurses, practice managers and educators delivering the intervention. The interviews and the group discussion were conducted by an experienced qualitative researcher (NA), who was not involved in development of the intervention. One-to-one interviews were conducted either face to face or over the telephone, according to participant preference, and lasted up to 30 min. All participants provided full written or audio-recorded consent prior to being interviewed.

### Data analysis

All interviews and group discussions were audiotaped and transcribed verbatim. Data were analysed thematically, informed by a constant comparison approach.[15] Analysis was undertaken concurrently with data generation, until saturation of themes. Data were compared within and between respondent groups (users, non-users and stakeholders), and between data from one-to-one interviews and group discussion for stakeholders. Data were reviewed, and emerging themes developed by both the

field researcher (NA), and another senior researcher (JK), with collective backgrounds in health psychology, health service research and clinical primary care. NA and JK were not involved in the development or implementation of the intervention.

## Patient and public involvement

The project approach was developed with local community members, including stakeholder meetings to develop and prioritise research questions and methods, in accordance with relevant guidelines.[16] This informed the qualitative approach, and included review of, and feedback on recruitment method, invitation letters, study information and consent forms by patients and other community stakeholders, with distribution of a summary of findings to all participants.

## FINDINGS
### Context

During the 12-month implementation period, 2053 patients from the nine general practices were identified at high risk of T2DM and invited to attend the programme. In total, 417 booked to attend (20% of those invited) with 369 individuals subsequently attending (16.5% of those invited) of whom two-thirds were aged over 60 years (mean 67.4 years, range 33–94 years), 52% were female, with over 76% of self-defined white-British ethnicity.

### Sample

Ultimately, 32 participants were purposefully sampled and interviewed (of 60 participants willing to be interviewed). This included 10 health professionals involved in local implementation (6 interviewed one to one, 4 participated in group discussion, including primary care practitioners, intervention facilitator/educators, commissioning leads) and 22 people at high risk of T2DM, summarised in table 1, who had either attended the *Let's Prevent* programme (n=15), or who had booked onto a session but did not attend or who declined the offer of intervention (n=7).

### Experience of diabetes prevention offer

Most participants had been surprised to receive the invitation from their general practice informing them that they were at high risk of T2DM (ie, they had 'prediabetes') and offering an intervention. These participants had not been previously aware of their risk and perceived insufficient involvement from their GP in relation to their referral for intervention (box 1). For some, their surprise at receiving an invitation provoked them to confront or try to make sense of their diagnosis of prediabetes. However, others had partly anticipated this from growing awareness of T2DM in the media or from personal experience such as their family history (box 1). Some had discussion of blood test results with their GP, and had expected the invitation to attend the intervention (box 1).

**Table 1** Characteristics of participants with elevated diabetes risk

|  | N=22 |
| --- | --- |
| Mean age in years (range) | 65 (41–83) |
| Gender |  |
| Female | 12 |
| Male | 10 |
| Self-defined ethnicity |  |
| White-British | 19 |
| South-Asian | 2 |
| Chinese | 1 |
| Educational level completed |  |
| No formal education | 4 |
| School | 7 |
| College | 3 |
| University | 7 |
| Existing long-term condition (eg, hypertension, osteoarthritis) | 16 |
| Self-reported current health promoting activities |  |
| Regularly active (eg, gym, sport) | 2 |
| Moderately active (eg, walking dog, gardening) | 11 |
| Inactive | 8 |
| Mean body mass index (range) | 28.7 (20.4–37.3) |

There were similar perceptions among most of those who had accepted and agreed to attend the intervention but who did not do so, and those who had simply declined the offer to attend. Some were apprehensive, having not had adequate information or follow-up of previous testing, or felt they were not a candidate for T2DM given a lack of symptoms. Others had only a vague awareness of their risk of developing T2DM from prior tests at their practice. They were unclear about the results from these tests or puzzled that these had not prompted any intervention thus far. In the context of their lack of preparedness, uncertainty or concerns, none of these participants opted to accept the invitation to attend the intervention or to actually attend having originally accepted the offer (box 1).

### Influences on uptake of intervention

Many who decided to attend the intervention education sessions appeared to be highly conscious and proactive about their health. For them, the decision to engage with the programme was an obvious choice (box 2). They anticipated the value of attending, particularly to gain, and be empowered by knowledge of how to reduce their risk of developing T2DM. Family history of diabetes also appeared an important motivation for attending.

Among those who had declined the intervention, several felt they did not want to worry about a condition that they had not yet developed, or highlighted other competing commitments or priorities such as their work, family, volunteering or other health appointments.

**Box 1  Experience of diabetes prevention offer**

**Surprised to learn of risk, with intervention offer:**

That is the first I knew about it, I had no idea that I was at any risk at all (…) I was a bit surprised, I thought it must be somebody else (…) nobody had ever mentioned it, you know, before. (Woman, aged 75 years, white-British, attended)

I was a little bit shocked because the doctor had given me no indication at all that I was getting to that stage of prediabetes (…) this came out of the blue. (Woman, aged 61 years, white-British, attended)

I would have liked to have been approached by the doctor before I ever got to that stage (…) the fact that the doctor had obviously got this information and had done nothing with it, on an individual basis with each patient. I think that was standard across the whole of the group [attending the intervention session] that none of them had been made aware by their doctors that they had entered this prediabetes stage. (Woman, aged 61 years, white-British, attended)

**Anticipated risk of diabetes:**

There is a big history of it in my family and my mother has it, my grandmother had it, type 2 and so you know I was half sort of expecting that possibly that it was something that I might get as well—although I have been I suppose a bit blasé about that knowledge that I might get it. (Woman, aged 61 years, white-British, attended)

I think I kind of expected it (…) my Mum is diabetic as well, type 2, so you know I have an awareness of diabetes, and you know the basic link to weight … And I realise that I was significantly over weight. So it just, you know, the penny hadn't dropped. Even though there were a lot of signals to tell me that I should do something. (Man, aged 42 years, white-British, attended)

It wasn't that I was going 'oh do you know I thought that was going to happen (…) I was just waiting for it'. On the other hand, I didn't fall through the floor with surprise. I mean there is plenty of coverage about the concern in rising diabetes so I just sort of thought 'oh fine'. (Man, aged 64 years, white-British, attended)

**Offer of intervention expected—prior awareness of diabetes risk:**

I knew that I had prediabetes as a result of, from getting results of previous tests (…) and that I was at risk of getting full blown diabetes (…) My GP talked to me about it. (Woman, aged 74 years, white-British, attended)

I went to the doctors for a blood test (…) That was when the doctor said your sugar levels are quite high, higher than normal (…) and he said 'we're doing a programme for prediabetics and they'll ring you up in due course'. (Woman, aged 49 years, South-Asian, attended)

She (GP) …probably said prediabetic (about blood results) but I didn't cotton on to that, and then I had this letter saying 'would I like to come?', and I said 'oh perhaps I am then'. And that's how I got to go to the course. (Woman, aged 63 years, white-British, attended)

**Response to intervention offer among non-attenders:**

I was scared (…) About all of it really… and what it meant and things like that (…) I have had a blood test … perhaps a couple of months back saying there was a lot of sugar … but it has not gone any further sort of thing. (Woman, aged 70 years, white-British, accepted intervention invitation, but did not attend)

I just received a letter from the surgery (…) inviting me to take part in a some sort of day because they said that I was at risk (…) I was not happy, in as much as I didn't believe them, that I was at risk and I still don't (…) if it is that I am at risk of this diabetes, why haven't I got any symptoms, feelings or anything else? (Woman, aged 71 years, white-British, declined intervention invitation)

A bit shocked really (…) it said because of the blood results … that I could be verging on the edge… it didn't say I have got diabetes (…) that was the first I knew (…) nobody had turned around to me and said 'oh by the way you have got diabetes'. (Man, aged 72 years, white-British, accepted invitation, but did not attend)

Others indicated they already had an awareness of T2DM, and were taking active steps in their lifestyle to reduce their risk of developing the condition. Some 'decliners' were further deterred by features of the programme itself, perceiving the intervention session seemed unnecessarily long for the group format. They would have preferred a more individualised approach (box 2).

**Information needs**

Participants' views about the information shared at the time of being offered the intervention were varied. Some attenders regarded the information in their invitation letter as sufficient to help them make an informed decision, or that any more information at that stage would have been unnecessarily anxiety-provoking. However, others felt that being told about their high risk of T2DM via letter suggested a problem that was not serious, and that a telephone or face-to-face consultation with the GP or being given a formal diagnosis of 'prediabetes' may have been more helpful (box 3). This mirrored views of some of those who declined the intervention (box 3).

Others were less satisfied with the information that they had received, but still attended the education sessions. These participants sought more prior individualised information about identification of their risk of developing T2DM, including their blood test results, more information on prevention of T2DM and on the specific content of the intervention (box 3). Others suggested that it would have been preferable to have been referred more directly, having seen their GP, rather than receiving a mailed invitation.

## Box 2   Influences on uptake of intervention

**Motivations for attending:**

If I was at risk then I better go and find out why I am at risk and what I should do about it. (Woman, aged 75 years, white-British, attended)

When you've got a problem, it's ridiculous to ignore it and just merrily carry on your own sweet way, without taking notes. (Man, aged 65 years, white-British, attended)

Well my family history. I don't want to be diabetic (…) so I was very happy to gain all the knowledge I could to prevent me becoming diabetic. (Woman, aged 63 years, white-British, attended)

I know quite a lot about diabetes but I thought I might learn something new (…) if I can prevent diabetes from say occurring in 2 years but if I get another 10 years without then yes it would be greatly benefit you know. (Man, aged 65 years, South-Asian, attended)

**Reasons for not engaging:**

I contacted them and apologised but I considered that my (other health problem) was more important (…) I am not going to start worrying about something I haven't got yet. (Woman, aged 83 years, white-British, accepted intervention invitation, but did not attend)

I am aware of type 2 diabetes, I try…to keep active, I try to eat a healthy diet, I certainly don't have lots and lots of fizzy drinks (but the) actual commitment to it (…) I don't work, but I have a lot of commitments (…) I didn't feel I could commit to something like that. (Woman, aged 66 years, white-British, declined intervention invitation)

The problem seems to be (it's) a whole day which to me seems very, very long (…) I thought 'no not worth bothering'…I couldn't understand why it takes so long, … from morning until afternoon (…) you switch off …by the time you have finished lunch your mind is not interested anymore so the rest of the afternoon session is just put to waste. (Man, aged 65 years, Chinese, declined intervention invitation)

I didn't feel that I needed a day …, you know with a load of other people, you know a group session, it is not my way. If…they feel that there is something wrong with me, I expect them to ring me and say 'I want to see you'. (Woman, aged 71 years, white-British, declined intervention invitation)

Some felt a lack of this range of information may have deterred other people from attending, and this was indeed underlined by those people who did not attend (box 3). While the nature of information required by non-attendees (more individualised rather than generic) was similar to those who attended the education sessions (box 3), it appeared unlikely that this would have altered peoples' decision about non-attendance.

### Intervention content, delivery and impact

Most attenders were generally positive about their education session. In particular, they enjoyed interaction with others; the resources provided (booklets and pedometers); dietary information and group format, which they found socially supportive, helpful and encouraging (box 4). They felt that they gained knowledge about T2DM that was relevant to their prediabetes status. These participants also felt that the educators were friendly, skilled and knowledgeable. They found booking onto the programme was easy, with flexible timing to choose their preference, including weekends. However, few sought a Saturday session, with only 4 of 48 programme sessions thus arranged on a Saturday. Other practical aspects of the programme that attendee participants liked included the convenience and accessibility of community venues, such as public library or a local community centre, in walking distance or with ease of parking (box 4). These attendee participants were largely retired, working part-time or self-employed, or did not work for health or other reasons, and found it straightforward to attend the education sessions.

However, attendees identified a range of challenges for the intervention. Several acknowledged it was attempting to cater for the needs of a broad range of people, within resource constraints, rather than providing more tailored individualised advice and guidance (box 4). Some found the sessions too general and untargeted. South-Asian participants, who had taken time off from work to attend, found their session did not cater to their cultural needs in relation to diet or feeling at ease in the session (box 4). In particular, they noticed a lack of other South-Asian attendees, and felt discomfort in a large group, preferring a less intimidating small group or individual one-on-one format.

Over half of those who attended sessions, including those who enjoyed them, felt that spending 6 hours was too long and onerous. They perceived there was a large volume of information that was repetitive, found the approach could be patronising at times or that the educational content lacked appropriate depth (box 4). Other practical concerns were identified. For example, several participants noted that while being offered a pedometer as part of the intervention was an advantage, they had found theirs was broken (box 4).

Most attendees felt that they had benefited and learnt something from the education session. They felt that it had enhanced their awareness of bad habits, and encouraged them to make lifestyle changes, particularly improving their diets, and for some participants, getting more exercise. Many found it helpful to learn more about the food 'traffic light' system, which they could use when buying food at the supermarket, although noted that they were already quite health conscious prior to attending (box 4). However, some perceived little gain. They felt the session had been superficial and not afforded practical help, or had provided broad information-overload at the expense of more focused guidance. Echoing earlier concerns about the process of referral and pre-intervention information, several participants were unclear about how to get further support or follow-up after the intervention, including their GP, and were uncertain about pursuing this for themselves (box 4).

### Stakeholder professional perspectives

Stakeholders implementing the intervention perceived the education sessions to be a useful resource, and the programme to be positively experienced by staff and

## Box 3  Information needs

**Adequacy of information with the intervention offer:**

I suppose I didn't take it as being that serious….I mean if you'd been given serious results, you know….you'd have perhaps preferred a face to face meeting. But as it was, it was more of a warning notice than anything else, and no I think it was handled fine. (Man, aged 65 years, white-British, attended)

I think there was sufficient information that—I think the main points were that, you know, this is my condition, that this course is going to help you deal with it. (Man, aged 42 years, white-British, attended)

I think the letter was OK. It's just that if they'd said I was 'prediabetic', it would have made more sense. (Woman, aged 64 years, white-British, attended)

I think what I had at the time was probably sufficient, because I think if they sent out a lot of the other stuff it would have probably rung even more alarm bells. (Woman, aged 61 years, white-British, attended)

**Desire for greater information with intervention offer—among attendees:**

I didn't receive any information (about) why I was being called (…) I think some people probably had the letter and didn't bother to go (…) I didn't know why I was going…why I was at high risk. (Woman, aged 75 years, white-British, attended)

They perhaps could have just a little bit more information…. on what you know what it is to prevent diabetes. (Woman, aged 48 years, white-British, attended)

It most probably would have been better for the doctor to have told me. I think that would have been better than just having the letter out of the blue. (Woman, aged 80 years, white-British, attended)

All it said was 'bring sandwiches, it's a whole day thing', and that was it really. And 'book up to attend' (…) I'd like a bit more on what to expect, yeah. I'd like a bit more information on what the programme is about, what the day is going to be, you know like an agenda. (Woman, aged 49 years, South-Asian, attended)

I would have preferred a telephone call to say come in and see the doctor (…) that could have been explained to me before I got there (intervention session), you know what the problem was. (Man, aged 68 years, white-British, attended)

**Insufficient information provided with offer—non-attendees:**

I couldn't understand what it is all about, I know yes it is 'diabetic' but you know (…) other than saying you start at 9 in the morning and finish about 4 in the afternoon, bring your own sandwiches … For (…) one whole day session, what is the content?, why take the whole day?, I need some explanation, at least then you can actually visualise why it takes all day. (Man, aged 65 years, declined intervention)

I would have preferred (information about) the symptoms and things like that and if it is hereditary … because I have got a big family because my son has got second diabetes, class two and …. you know what chance is it of me getting it. Some sort of information like that you know (…) (Woman, aged 70 years, white-British, agreed but did not attend)

If they had given me my personal information 'we feel you're at risk because of this, this and this', I would have taken it seriously. But it looked to me like it was a standard letter that was sent out to everybody, well we can't all be the same (…) I would have liked them to have given me the reasons that they had sent me the letter. (Woman, aged 71 years, white-British, declined)

It probably would have been helpful (if approached by general practice) but it still wouldn't have changed the circumstances (attended or not). (Man, aged 72 years, white-British, Agreed but did not attend)

patients. However, some primary care stakeholders questioned feasibility and sustainability because of limited time and financial constraints, for staff and for patient engagement. They suggested other less time-consuming and potentially cost-effective ways to deliver prevention information to those at risk were also needed, including audio-visual or online resources. Stakeholders further debated a tension between adopting a locality-wide model of primary care to deliver sessions in the community as happened here, or a model of intervention facilitation within individual general practices themselves as part of routine care, or achieving an appropriate balance between the two. Some suggested the principal reason the intervention was implemented was because the GP Federation facilitated the identification of eligible participants and organised the invitations on behalf of practices, using a locality-based coordinator, which reduced the workload of busy general practice teams. In contrast, others felt resourcing and initiating more of this work,

for example, with a designated individual, within each general practice would have been preferable, to increase patient engagement and awareness of their elevated risk, to facilitate referral to the intervention (box 5).

Reflecting on the intervention programme, stakeholders identified several issues that had not been actively anticipated. This included the generally low awareness patients had of their high risk of T2DM, how this may have affected engagement and the need for this to be improved before referral or with offer of intervention. They underlined the absence of clear prevention pathways hereto for people they identified at high risk of T2DM, and the wider ethical challenge this posed (box 5). Stakeholders felt this was an issue that general practices were increasingly aware of, and trying to address. Similarly, they identified the importance of follow-up and continuity of support for prevention, with potential for general practices to undertake this and monitor patients to capitalise on investment in the intervention (box 5).

## Box 4  Intervention content, delivery and impact

**Positive aspects—interaction and information:**

I thought it was very good (…) they were able to answer questions…. And they also told us things that we weren't aware of (…) They didn't try and put doom and gloom, or force it down your throat. They just gave you the facts and they put in a fairly, I'd say light hearted, more friendly fashion. (Man, aged 65 years, white-British, attended)

It was good, because you heard other people's stories. (Man, aged 65 years, white-British, attended)

I didn't really know … there is too much glucose in the blood and the pancreas struggles (….) And so I learnt all about that with the others which was very interesting and very helpful. (Man, aged 71 years, white-British, attended)

**Practical convenience and accessibility:**

Being nearly retired getting the time off to do it wasn't a problem and there was a number of venues that I could pick from…, at the most appropriate day. And that was good. (Man, aged 71 years, white-British, attended)

It was right beside … my doctors. So I knew where it was. And it was a very nice place, a venue to go to. (Woman, aged 63 years, white-British, attended)

The venue was local, I only work part-time, so that wasn't an issue. (Man, aged 68 years, white-British, attended)

**Challenges of generic programme and cultural relevance:**

There were obviously people with a range of circumstances (…) they sort of started off with some generalities, there wasn't an assumption about people's level of knowledge (…) Until the group assembles, the (facilitators) actually have no idea how receptive their potential audience is going to be. (Man, aged 64 years, white-British, attended)

I don't suppose they have got time to get every patient in and discuss their own (individual) issues… (Man, aged 68 years, white-British, attended)

It was daunting at first (…) it was about 20 odd (…) it was quite a large group. I would have preferred a smaller group (…) or an individual session (…) Obviously with finances and resources, they can't do that. It's got to be a general thing (…) but obviously with my ethnic background, …(and) being vegetarian as well, I asked a question about the carbs …. and how do I increase the protein intake, and they really couldn't answer any of those questions. …Exercise I get, but the food wise, doing curries and that, how do I adapt? (…) it wasn't for me, you know, it wasn't, for my background. (Woman, aged 49 years, South-Asian, attended)

I don't think it refers to Asian(s), you know the thing is mainly geared for the English community (…) the majority of them (attending) were English people ….I was the only Asian. (Man, aged 65 years, South-Asian, attended)

**Challenges for intervention delivery—length and depth of session:**

It needn't have lasted all day… (it) could have been condensed in to about 2 hours (…) it was far too long (…) we just kept going over the same things. Repeating itself. It was like being at infant school, …. (…) a bit condescending really. (Man, aged 47 years, white-British, attended)

It was a little bit slow (…) there was kind of a bit of a waffle in it we could have done it probably in half a day (…) I found it a bit simple …. too dumbed down for me. (Woman, aged 75 years, white-British, attended)

If it had run on much longer, it would have seemed that we were regressing to kindergarten. (Man, aged 73 years, white-British, attended)

They briefly touched on everything (….) but (the programme should) go into a bit more depth … (Woman, aged 49 years, South-Asian, attended)

Other practical concerns:

They gave us one of these pedometer things but … you know it is just a sort of very basic one. (Woman, aged 75 years, white-British, attended)

I'm not convinced the pedometer's accurate. (Woman, aged 74 years, white-British, attended)

Because I work full time, I had to book a locum … so that I could have a day off, … so it did cost me quite a bit of money you know because it (intervention session) was on a week day. (Man, aged 65 years, South-Asian, attended)

**Positive impacts of intervention:**

A lot of people didn't realise about these traffic lights on food—you know—I think they are a good thing (…) I have definitely started to eat a bit more fruit, again. (Woman, aged 48 years, white-British, attended)

I learnt quite a few things from it (…) We are eating a lot more sensibly now… and … well we have always walked for miles every week. (Man, aged 47 years, white-British, attended)

It was a bit of a wakeup call (…), it probably wouldn't have dawned on me that it (risk of diabetes) was an issue. (Man, aged 64 years, white-British, attended)

**Impact of attendance—concerns:**

I think the whole thing was a tick box exercise to say 'right, (….) we've covered it, we've told them that they're pre diabetic, get on with it' (…) I've not come out with anything really (…) if I'd known that, … I don't think I would have bothered (…) I could have just Googled it. (Woman, aged 49 years, South-Asian, attended)

It would have been (useful) if you could take it all in. (Woman, aged 80 years, white-British, attended)

One of the (educators) said after 3 months you can go and have another HbA1c done and then he said it might show a bit of difference—but nobody has contacted me. (Man, aged 65 years, South-Asian, attended)

## Box 5   Stakeholder professionals' perspectives

### Approaches to supporting intervention:

The thing that made it work and the reason it did work was being able to employ the project coordinator to pick up the work on behalf of the practices. So they weren't asking practices to do any additional work that took them away from their normal day to day resource. (…) because general practice is so stretched. (General practice Manager)

I don't think it's sustainable as it is, not with a whole day programme (…) the message is very important, but I think delivering it in the current format is not really achievable under current financial constraints and also constraints for the patient in terms of time (…) I mean one way would be a… web based video …to see this programme delivered in a shorter space of time, without actually attending. (General practitioner)

The referral pathway that would be ideal was the opportunistic one where the patients (see) the GP or the healthcare professional and then are being asked and being referred on, just didn't happen. (Project facilitator 2)

I would (….) maybe fund somebody within a practice for a small number of hours… I think one of the barriers maybe people didn't quite understand who was contacting them and why and where that fitted in (…) If the receptionist (they) know or who ever contacts them, I think it gives (…) reassurance it is something worth doing and it feels more it came from your GP (…) (General practice Manager)

I think the reason why people didn't take part, whether it was because it wasn't practice based, or the main barrier was time, or whether they didn't understand quite how Let's Prevent fitted in with practices. (General practice Manager)

### Anticipating challenges for patient engagement:

So one of the first questions that you ask people [at the education session] is 'how did you find out you had prediabetes?' and most people, the vast majority … said that 'I didn't know until I got a letter inviting me to come here' I think 90% of people said that. (Intervention educator).

There were some that were really quite upset nobody had ever told them and they only found out by the letter …they were considered prediabetic. (Project facilitator 1).

You've got a register of people (with prediabetes) ….then obviously something like this (intervention) comes along and you're pulling patients off, almost with the assumption that patients know they're already on that register. (GP federation stakeholder).

It's slightly unethical really (…) To actually identify somebody at risk of something, it's like having a genetic test, for whatever and not being told the results (…) it's like we know and you don't. (Intervention educator)

The main thing is to make sure that going forward that people who did attend are monitored (…) for a change in their outcome. (General practice Manager).

## DISCUSSION
### Principal findings

This study suggests the *Let's Prevent* pragmatic diabetes prevention intervention, implemented in community practice, is acceptable and could be helpful for those choosing and able to attend. Respondents were most positive about supportive interaction with others, educator expertise, dietary advice and local accessibility. Those who engaged tended to be older, white, often retired and with time to attend, who were motivated to and reported making changes to their lifestyle. While the challenge of offering education suitable for all was recognised, some experiences of the intervention were perceived more negatively. These included lack of depth or cultural relevance in content, finding the session overlong or repetitive, doubts about quality of pedometers provided and absence of follow-up after the intervention.

Key factors hindering engagement, with potential for harm, were lack of preparedness for the intervention offer, with low prior awareness of elevated diabetes risk. Most had little preceding communication about this. In particular, those who defaulted from, or who declined the intervention were apprehensive, uncertain or unconvinced about whether they were at risk of diabetes. They sought more specific information about intervention content, and were deterred by the day-long commitment, group format or competing obligations to work and family. Local providers welcomed the opportunity for prevention offered, but highlighted challenges for engagement, such as communication of risk to patients, and for implementation.

### Strengths and limitations

This study provides insights into how implementation of a pragmatic diabetes prevention programme was experienced in real-life practice, suggesting how patient engagement might be enhanced, and aspects of the intervention that might be developed. The purposeful sample reflects the demographic age, gender and ethnic profile of the population who had the intervention, and local stakeholders directly involved. A particular strength is inclusion of those who declined the intervention offer or who accepted but then defaulted from attendance. Other strengths are that data were generated and analysed by two researchers of different disciplinary backgrounds, who were not involved in developing or implementing the *Let's Prevent* intervention itself. In addition, stakeholder input to study conduct was gained throughout its development, in addition to plans for implementation of the *Let's Prevent* intervention itself.

Limitations of this work should be noted. The ethnic composition of our patient sample reflected that of the population opting to receive the intervention, which was almost 80% white. In practice, this constrained the wider ethnic diversity of those willing to be interviewed. We successfully included people with a range of educational levels, and South-Asian and Chinese-origin respondents in our sample but acknowledge ethnic diversity was limited. Quantitative evaluation is also needed to help assess feasibility, with process and outcome data for the whole population approached for intervention, and this is being undertaken and reported separately. This included reasons for declining the intervention recorded for 169 people and the current findings are consistent with these.

Our qualitative findings should be regarded with respect to the study sample and context described.

## Relation to other studies

Recent evaluation of the first national wave of the NHS Diabetes Prevention Programme across seven demonstrator sites in 2016 reviewed service procedures for the specified intensive interventions (using structured group education, similar to *Let's Prevent*, but delivered to groups of 15–20 adults in at least 13 sessions over 9 months). That review included qualitative research with stakeholders and 21 service-users,[17] identifying or anticipating issues consistent with the current findings. Stakeholders highlighted concerns for patient referral and uptake, the role of primary care in supporting this, long-term sustainability, intervention reach and equity. Service users reported benefit from social support of peers, and the challenges and opportunities of modifying diet. The current study adds to this work by offering experiences of a diabetes prevention intervention in more detail, by also including perspectives of those who defaulted from or declined intervention and by providing insights into the use of a more pragmatic single-session intervention. This may be of particular relevance given lower completion rates in similar US diabetes prevention programmes with increasing number or length of intervention sessions.[18]

Current evidence[19] suggests the delivery and content of public health educational interventions are necessarily highly variable. While interventions may improve knowledge, skills, self-efficacy, attitudes or behaviour, for example, firmer evidence providing clearer understanding of how such changes may occur is still needed. The change approach used in the *Let's Prevent* intervention is summarised elsewhere.[8–11] Recent work advocates more integrated approaches, for example, combining a Capabilities, Opportunities, Motivation-Behaviour model[20] with behavioural insights (such as the Messenger, Incentives, Norms, Defaults, Salience, Priming, Affect, Commitment, Ego (MINDSPACE) framework[21]) to promote behaviour change in the NHS Diabetes Prevention Programme.[22]

## Implications for practice, policy and future research

Our findings underline that structured group education intervention for diabetes prevention may work well for participants who engage with and complete intervention sessions, for example, those who are older, more health conscious and from less deprived and white-European backgrounds. However, significant challenges for reaching those from socially deprived and ethnically diverse communities remain, with concerns for equity and impact, in addition to resource-related long-term sustainability.[17] This remains true in other contexts such as diabetes screening.[23]

The study emphasises how evidence accrued in a formal trial, with consenting randomised participants,[8 10 12] can differ from real-life experience (despite attempts made to culturally tailor intervention content). One intervention

model is unlikely to engage all those at high risk or be effective for all, and greater attention to the local socio-demographic context is needed. Failure to reach younger people of working age under 60 years, and in particular those from more deprived and minority ethnic communities, at highest elevated risk of diabetes, risks further perpetuating health inequalities. Cultural adaptation of interventions, with appropriate community support for engagement and delivery, tailored for local communities may help.[24] This approach has shown promise, for example, with UK Pakistani women.[25] This might include 'generic' and designated sessions for specific groups, which was not possible in the current implementation.

Respondents' experiences point to specific considerations for further *Let's Prevent* development, and expose the tension between using a pragmatic lower intensity intervention or offering more resource-intensive multiple sessions over time. A single 6-hour session was felt over-long or was difficult to commit to, yet greater depth of content, more culturally specific content and follow-up would have been welcomed. Replacing some of the time-intensive face-to-face contact of traditional intervention approaches by exploiting online, smart-phone or other digital technologies are promising possibilities,[26] as some stakeholders suggested here, and might also engage younger people and those in work.

Users' lack of awareness or confusion about elevated risk and lack of preparedness for the intervention offer were major issues for engagement and uptake, with potential to cause anxiety and uncertainty. This highlights the importance of improving effective communication of raised diabetes risk prior to, and as part of intervention referral, to increase engagement and avoid potential harm. Both user respondents and primary care stakeholders noted better mechanisms needed to be in place for this to happen to ensure more appropriate care pathways. This should include when elevated diabetes risk is identified by NHS Health Checks. More direct individualised communication by GPs may be preferable, as some users suggested, in addition to adequate specific information accompanying the intervention offer, and active primary care follow-up after the intervention. More specific general practice phone contact to patients at any of these stages might also be considered.

With largely locality-wide facilitation of implementation here, no costs were borne by general practices themselves, but stakeholders debated what might be better done at individual practice level versus using a locality approach. Achieving an appropriate balance between the two will be important for diabetes prevention interventions, linked to appropriate care pathways for those approached, and underlines the importance of linking services within local systems for this purpose (such as CCG, local authority public health and leisure, primary care general practices). While stakeholders were positive about this intervention overall, they questioned sustainability given constraints on time, staff capacity and future

resource allocation, presenting a challenge for diabetes prevention programmes going forward.[19]

## CONCLUSION

This pragmatic diabetes prevention intervention introduced into practice was acceptable, particularly for older and health-motivated people with time to attend. Further development of this, or other interventions is needed to enable wider and more socially diverse engagement of people at risk, and to avoid perpetuating health inequalities. Better pre-intervention information and effective communication of diabetes risk are required, with closer integration of services to facilitate engagement, uptake and follow-up.

**Acknowledgements** The authors would like to thank all interviewees and others who supported this project, including University of Nottingham as study sponsor, North Charnwood GP Federation, the *Let's Prevent* project implementation group, including Pat Drinkwater for assistance with participant recruitment and Naina Patel for qualitative protocol development and PPI work, the interview transcribers and Pamela Pepper for administrative research support at the Division of Primary Care, University of Nottingham.

**Contributors** NA carried out fieldwork and data generation supervised by JK, who was principal investigator. NA and JK analysed the data, and wrote the paper. HD and TY supported liaison with the implementation project, and helped revise the paper. All authors reviewed and approved the final draft of the paper.

**Funding** This study was funded by the Division of Primary Care, University of Nottingham and NIHR CLAHRC East Midlands. Implementation of the intervention was supported by NIHR CLAHRC East Midlands and the East Midlands Academic Health Science Network (AHSN).

**Competing interests** NA, JK have no conflicts of interest to declare. TY and HD were involved in intervention implementation as part of the *Let's Prevent* team. TY has also contributed to the design of a diabetes prevention programme run through an NHS (University Hospitals of Leicester NHS Trust) and industry (Ingeus UK Limited) collaboration which is part of the tendering framework for Healthier You: The NHS Diabetes Prevention Service.

**Patient consent for publication** Not required.

**Ethics approval** Ethical approval was obtained from University of Nottingham Faculty of Medicine and Health Sciences Research Ethics Committee (reference: A15032016).

**Provenance and peer review** Not commissioned; externally peer reviewed.

**Data availability statement** Data are available from the authors.

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
