## [Reviewer comments · BMJ Open]

ARTICLE DETAILS

TITLE (PROVISIONAL)	Users' experiences of a pragmatic diabetes prevention intervention implemented in primary care: Qualitative study
AUTHORS	Aujla, Navneet; Yates, Thomas; Dallosso, Helen; Kai, Joe

VERSION 1 – REVIEW

REVIEWER	Jorg Huber University of Brighton, UK
REVIEW RETURNED	07-Feb-2019

GENERAL COMMENTS	This is a nice paper. Well argued, written and structured. I have one main concern: the paper is somewhat narrowly focused which is a strength, but it is also a limitation as it does not address the more conceptual or theoretical literature around behaviour change and structured education programmes. Nor does it attempt to generalise. These issues make it of limited importance to other fields, outside of diabetes. Nor will it be of wider interest to those developing behaviour change programmes, aiming at prevention, whether primary or secondary, more widely. If the authors could try to highlight some elements as suggested above, without lengthening the paper unnecessarily, it might become of considerably greater interest to a wider audience. One thing which could be shortened is the the initial paragraph. It may be personal fatigue, but the mantra like declaration of the prevalence, burden and costs around diabetes are becoming very tiresome, and many people will know this by now. Directing the introduction more on the qualitative study which is reported would again make it more interesting. Thinking about this study further, it reads more like a case study, although I can clearly see its wider relevance and its importance in relation to efforts to reduce progression to full diabetes, in at risk populations.
---

REVIEWER	Katherine Murphy Chronic Disease Initiative for Africa, Department of Medicine, University of Cape Town, South Africa
REVIEW RETURNED	11-Mar-2019

GENERAL COMMENTS	I would recommend that this study be accepted for publication with a few minor revisions/additions. The study has a strong rationale, its methodology is sound and its findings add meaningfully to the knowledge base. It is very well written: clear and accessible without any excessive wordiness or unnecessary verbiage. The authors' arguments are cogently presented, and their recommendations are compelling. The quotes aptly illustrate the findings and are interesting and rich in detail. Developing diabetes prevention programmes which are acceptable and appealing to the target population, feasible to implement and effective in changing lifestyle behaviours and diabetes risk remains challenging and complex. This study is highly relevant to those charged with the responsibility of developing such interventions and to those who formulate related public health policy and strategy. I would like to make a few suggestions. I also have some queries... Queries: Pg 6 line 56: how were the materials tailored to the local population? I would be interested in more detail here. Pg 7 line 17: here it says the facilitators were trained in a 2-day standardised programme, but it is clear that this intervention was delivered in one day over 6 hours. So, can one presume that a 2-day programme was condensed into one day? If so perhaps it is no surprise that some people felt there was information overload... Pg 9 line 47: why, if a total of 60 people were willing to be interviewed, were so few people who did not attend interviewed (7 v 15)? What % of non-attenders were willing to be interviewed? Seeing as reasons for non-attendance was such an important research question, I would have thought that at least the same, if not more, non-attenders and defaulters (and participants from minority communities) should have been included in the purposive sample. Weighting the sample in favour of non-attenders rather, could have offered more insights from a "wider and more socially diverse group" ie. those who are from minorities and more disadvantaged groups. The greater challenge is how to reach and appeal to these groups. I do, however, appreciate that this is noted in the conclusion on page 25 as something needing further research. Pg 17: in the methods it is stipulated that 6-10 people were recruited to a group, but in the quote on line 34, the participant says that her group consisted of 20 people. Was this a departure from the intervention protocol? Pg 17: Challenges: I found these quotes interesting and they raised a question for me as to what approach was used in the programme. From these quotes, it seems to me that the session might have been very 'information heavy' and perhaps delivered in a lecture style, with few or no interactive activities for shared problem solving or to teach/demonstrate practical behaviour change skills. Was this the case? On page 5 there is a description of the content of the intervention, but not the approach used in the delivery or the communication style. So, for eg how was self-efficacy and goal setting actually addressed in the programme? Were participants engaged in exercises or activities which involved them in a meaningful and personal way? Or was the approach predominantly information and advice-giving? I would recommend that this aspect be more fully described. Suggestions:
--

	Pg 8 line 56: the fact that the study and implementation plan for the intervention was developed with input from stakeholders is a strength that should be highlighted. Strengths and limitations are otherwise comprehensive. Pag 22: Relation to other studies: I would suggest that a few more references be added here from the general literature on these kinds of interventions. What is most recent, quality evidence on the content, mode of delivery, approach to education and counselling and intervention dose? What seem to be the essential elements of effective diabetes education programmes which have the most evidence to back them up? This would give greater context for the findings of this study. Pg 23 Does any literature show that a greater proportion of typical non-attenders tend to be from minority or more disadvantaged communities? That retired, better off, self-identifying white, British people are the most likely to attend such programmes in the UK? Was this the case in the trial? A discussion of this issue and the reasons why this might be the case in the Lit review and/or discussion section (pg 23, 2nd paragraph) would be instructive. The issue of feelings of cultural alienation seem very important... has this been found in other local studies and is it usual practice to have separate sessions for different cultures in the UK? This surely would come with its own problems as it could be seen as separating people/ not treating them the same??? I would like more elaboration on these kinds of issues on page 23 lines 31-36 and perhaps reference to another couple of studies.
--	---

VERSION 1 – AUTHOR RESPONSE

REVIEWER 1 COMMENTS	AUTHOR RESPONSE
1. This is a nice paper. Well argued, written and structured. I have one main concern: the paper is somewhat narrowly focused which is a strength, but it is also a limitation as it does not address the more conceptual or theoretical literature around behaviour change and structured education programmes. Nor does it attempt to generalise. These issues make it of limited importance to other fields, outside of diabetes. Nor will it be of wider interest to those developing behaviour change programmes, aiming at prevention, whether primary or secondary, more widely. If the authors could try to highlight some elements as suggested above, without lengthening the paper unnecessarily, it might become of considerably greater interest to a wider audience.	Many thanks for this positive assessment and feedback. We agree the focus of the study on this intervention, in this context for diabetes prevention, is a strength. In the Discussion we have now highlighted further reference to literature on structured educational interventions and behaviour change to widen interest. This has been done so as not to overly lengthen the paper - as kindly suggested. See Page 23, Lines 624-631. (This addition has also been done in response to Reviewer 2’s similar comment no. 8 below).
2. One thing which could be shortened is the initial paragraph. It may be personal fatigue, but the mantra like declaration of the prevalence, burden and costs around	We agree with this comment. We have considerably shortened the first paragraph in the Introduction accordingly, as indicated on p5, Lines 16-31.

diabetes are becoming very tiresome, and many people will know this by now. Directing the introduction more on the qualitative study which is reported would again make it more interesting.	
3. Thinking about this study further, it reads more like a case study, although I can clearly see its wider relevance and its importance in relation to efforts to reduce progression to full diabetes, in at risk populations.	We agree this can also be read as a case study.
REVIEWER 2 COMMENTS	AUTHOR RESPONSE
1. I would recommend that this study be accepted for publication with a few minor revisions/additions. The study has a strong rationale, its methodology is sound and its findings add meaningfully to the knowledge base. It is very well written: clear and accessible without any excessive wordiness or unnecessary verbiage. The authors' arguments are cogently presented, and their recommendations are compelling. The quotes aptly illustrate the findings and are interesting and rich in detail. Developing diabetes prevention programmes which are acceptable and appealing to the target population, feasible to implement and effective in changing lifestyle behaviours and diabetes risk remains challenging and complex. This study is highly relevant to those charged with the responsibility of developing such interventions and to those who formulate related public health policy and strategy. I would like to make a few suggestions. I also have some queries...	Many thanks for such positive feedback on our paper. We are glad the study is deemed methodologically sound and is found compelling.
2. Pg 6 line 56: how were the materials tailored to the local population? I would be interested in more detail here.	An additional sentence on this has been added on Page 7, Lines 65 and 67.
3. Pg 7 line 17: here it says the facilitators were trained in a 2-day standardised programme, but it is clear that this intervention was delivered in one day over 6 hours. So, can one presume that a 2-day programme was condensed into one day? If so perhaps it is no surprise that some people felt there was information overload...	The training for the facilitators by the Leicester diabetes centre was a 2 day programme for trainers. This has been clarified in the text (Page 7, Line 64). This was distinct from the intended one day intervention itself.
4. Pg 9 line 47: why, if a total of 60 people were willing to be interviewed, were so few people who did not attend interviewed (7 v 15)? What % of non-attenders were willing to be interviewed? Seeing as reasons for non-attendance was such an important research question, I would have thought that at least the same, if not more, non-attenders and defaulters (and participants from minority communities) should have been included in the purposive sample. Weighting the sample in favour of non-attenders rather, could have offered more insights from a "wider and more socially diverse group" ie. those who are from minorities and more disadvantaged	We agree that interviewing even more of those who were unwilling to engage with the intervention or who defaulted would indeed be ideal. However the study experienced the common challenge of being restricted to sampling those willing to be interviewed - and the large majority of these were those who had also engaged with the intervention itself. Considerable and successful efforts were thus made to include people who had not engaged with the intervention, but were nevertheless willing to be interviewed about why. This ultimately formed a third of the relevant sample.

groups. The greater challenge is how to reach and appeal to these groups. I do, however, appreciate that this is noted in the conclusion on page 25 as something needing further research.	(Their more detailed views, as described, were also noted to accord with responses from 169 other people indicating their reasons for not attending when replying to intervention invitation. This is noted in the Discussion Page 22, Lines 601-605, referring to a forthcoming quantitative report). As kindly recognised by the reviewer, we note in the Discussion that aspects of study participants' social diversity were more limited, and the wider challenge of this for similar interventions and research (Page 21, Lines 597-601).
5. Pg 17: in the methods it is stipulated that 6-10 people were recruited to a group, but in the quote on line 34, the participant says that her group consisted of 20 people. Was this a departure from the intervention protocol?	We believe this to be the case. The intervention itself was being implemented in the reality of day to day practice in the community, rather than as formal research itself (the preceding formal intervention trial is referred to in the Introduction and Discussion).
6. Pg 17: Challenges: I found these quotes interesting and they raised a question for me as to what approach was used in the programme. From these quotes, it seems to me that the session might have been very 'information heavy' and perhaps delivered in a lecture style, with few or no interactive activities for shared problem solving or to teach/demonstrate practical behaviour change skills. Was this the case? On page 5 there is a description of the content of the intervention, but not the approach used in the delivery or the communication style. So, for eg how was self-efficacy and goal setting actually addressed in the programme? Were participants engaged in exercises or activities which involved them in a meaningful and personal way? Or was the approach predominantly information and advice-giving? I would recommend that this aspect be more fully described.	We agree with this interpretation. We have added further information on the intended delivery approach for the intervention on Page 5, Lines 35-37. As this reviewer rightly notes from the findings presented, the reality of participants' experience of the intervention implemented in practice was that it was information-predominant.
7. Suggestions: Pg 8 line 56: the fact that the study and implementation plan for the intervention was developed with input from stakeholders is a strength that should be highlighted. Strengths and limitations are otherwise comprehensive.	Thank you for this comment. We have now further highlighted this in the Discussion under Strengths and Limitations (additional sentence Page 21, Lines 593-595).
8. Pg 22: Relation to other studies: I would suggest that a few more references be added here from the general literature on these kinds of interventions. What is most recent, quality evidence on the content, mode of delivery, approach to education and counselling and intervention dose? What seem to be the essential elements of effective diabetes education programmes which have the most evidence to back them up? This would give greater context for the findings of this study.	Thank you for this suggestion. Reviewer 1 also suggested brief further reference to literature on structured education of the type used, and other research to contextualise. We have added this in the Discussion on Page 23, Lines 622-631.

9. Pg 23 Does any literature show that a greater proportion of typical non-attenders tend to be from minority or more disadvantaged communities? That retired, better off, self-identifying white, British people are the most likely to attend such programmes in the UK? Was this the case in the trial? A discussion of this issue and the reasons why this might be the case in the Lit review and/or discussion section (pg 23, 2nd paragraph) would be instructive. The issue of feelings of cultural alienation seem very important... has this been found in other local studies and is it usual practice to have separate sessions for different cultures in the UK? This surely would come with its own problems as it could be seen as separating people/ not treating them the same??? I would like more elaboration on these kinds of issues on page 23 lines 31-36 and perhaps reference to another couple of studies.	Thank you for this comment. The reviewer is correct that in the preceding formal trial of this intervention (and similar programmes) it is older, retired, white and more affluent people most likely to attend. This experience in the preceding trial and the related challenges of cross-cultural engagement and adaptation are further discussed in two paragraphs of the Discussion (under Implications for practice, policy, future research Pages 23 and 24, Lines 634-652), also highlighting relevant existing research (references 12, 14, 19, 27, 28). We have also made an additional reference on Page 23, Line 639-640 (citation number 25) to a UK study of diabetes screening with similarly disproportionate uptake by retired, white middle class people compared to others.
---	---

VERSION 2 – REVIEW

REVIEWER	Jörg Huber University of Brighton, Brighton, UK
REVIEW RETURNED	01-May-2019

GENERAL COMMENTS	At times, in the abstract and in the intro, the paper reads a bit like a quantitative paper when it was purely qual. Also the order of statements in the Results of the abstract could be improved, generally improving the structure and identifying key themes or topics/concerns. It would be good to know what method of qualitative analysis was used. I am also a little concerned about the interpretation of the findings. To me there are serious concerns, expressed about the communication of risk, the lack of fully-developed pathways of care/support for patients (the latter expressed by health staff). Coming from an NIHR perspective, the study does not really assess the feasibility of the intervention. This would require statistics about numbers of offers, acceptances and levels of attendance. This is obviously not the purpose of this paper; so such confusions should be avoided, as the most likely reader is from England/UK. One of the strengths mentioned of independence of the researchers from the intervention developers is not - I cannot find a statement to this effect under sampling and data generation. Details of data analysis and PPI. Could a reference be provided? How did PPI shape the study? As it stands, there is little detail on the latter and it reads a bit like a box-ticking exercise (sorry). The strength and limitation section should probably report the main findings.
---

	In my view the paper requires attention above all in  - improving the research question/aim (I suggest to remove the notion of feasibility as this has now a very specific meaning in the UK); a term needs to be found whether it is suitable for roll-out which is a concern for some of the health staff - adding some details on ethics and generally improving the methods section - results are generally fine; but: at times it would have been good for a study like this whether some issues were mentioned more commonly than others (in particular the diabetes worries as I now call should be addressed and an indication of whether all, most, some or just a few did so; I know some qualitative researchers do not like this, but it is becoming more common to see such qualifiers in qualitative medical/health papers). - in the discussion and abstract the issues of harm (i.e. people being worried about diabetes risk and what it means) and the lack of care pathways needs to be highlighted more. In fact this is perhaps the most important finding, but would require some further detailed comments as suggested. As I added a few comments to the text, I add this. But these are just a few. I recommend that the paper is read by another person/or more attention is given to details of wording etc.. The reviewer provided a marked copy with additional comments. Please contact the publisher for full details.
--	--

VERSION 2 – AUTHOR RESPONSE

REVIEWER 1 FURTHER COMMENTS (May 2019)	AUTHOR RESPONSE (17 June 2019)
	We note Reviewer 1's earlier positive assessment of the paper with two helpful suggestions (March, above). Responses to this reviewer's further new comments are below. New changes in the manuscript are highlighted in blue (with previous changes in yellow).
1. At times, in the abstract and in the intro, the paper reads a bit like a quantitative paper when it was purely qual. Also the order of statements in the Results of the abstract could be improved, generally improving the structure and identifying key themes or topics/concerns. It would be good to know what method of qualitative analysis was used.	The abstract indicates the design is a 'qualitative interview-based study' and the Introduction states 'In this qualitative study' preceding the study aim. The abstract has now been slightly revised to lend emphasis to key themes. The method of analysis is detailed in the data analysis section of Methods but is also now stated in the abstract.
2. I am also a little concerned about the interpretation of the findings. To me there are serious concerns, expressed about the communication of risk, the lack of fully-developed pathways of care/support for patients (the latter expressed by health staff). Coming from an NIHR perspective, the study does	We agree the experiences reported in the study reflect some significant concerns. We appreciate assessment of feasibility requires both qualitative and quantitative methods. This paper sought to explore user experiences

not really assess the feasibility of the intervention. This would require statistics about numbers of offers, acceptances and levels of attendance. This is obviously not the purpose of this paper; so such confusions should be avoided, as the most likely reader is from England/UK.	qualitatively, thus including aspects of feasibility in practice. We agree it has not fully assessed feasibility, nor was it intended to do so. To avoid any potential confusion we have removed all reference to 'feasibility or feasible' from the paper (including Abstract and in Discussion). As the reviewer recognises, reporting of quantitative data was not the purpose of this paper. The future reporting of parallel quantitative data required for this project (including the type of data the reviewer refers to) is twice referred to in the paper (in the bullet pointed strengths and limitations section at the start of the paper, and in the Discussion under strengths and limitations).
3. One of the strengths mentioned of independence of the researchers from the intervention developers is not - I cannot find a statement to this effect under sampling and data generation.	This is referred to in the Discussion (strengths and limitations), and noted in COI section at the end of the paper. We have now also amended the Methods section to explicitly state the two independent researchers (NA/JK) undertaking sampling and data generation were not involved in the intervention development or its implementation.
4. Details of data analysis and PPI. Could a reference be provided? How did PPI shape the study? As it stands, there is little detail on the latter and it reads a bit like a box-ticking exercise (sorry).	Two further references to method of data analysis, and to PPI method have now been added. The PPI section in the paper is brief and as instructed in BMJ Open guidance for this section. PPI shaped the study in the ways described here. The paper has exceeded recommended word limit following incorporation of a previous main suggestion (April) constraining addition of further detail.
5. The strength and limitation section should probably report the main findings.	BMJ Open guidance instructs this section (at the beginning of the paper) should only refer to Methods and not include Findings. Hence main findings are not referred to here. Otherwise, in addition to the Findings itself, the main findings are summarised at the beginning of the Discussion.
6. In my view the paper requires attention above all in a) Improving the research question/aim (I suggest to remove the notion of feasibility as this has now a very specific meaning in the UK); a term needs to be found whether it is suitable for roll-out which is a concern	As noted above, we have now removed reference to feasibility or feasible from the paper. The main study aim refers to exploring the experiences of users, and the perceived value and acceptability of

for some of the health staff	the intervention in practice.
b) Adding some details on ethics and generally improving the methods section	Details of ethics approval were provided at the end of the paper, and are now also added in the Methods section. As noted above, we have also added independence of researchers and further reference to data analysis in Methods. We appreciate Reviewer 2 (above) deemed the study methodologically sound, and the paper clear and accessible by avoiding verbosity.
c) Results are generally fine; but: at times it would have been good for a study like this whether some issues were mentioned more commonly than others (in particular the diabetes worries as I now call should be addressed and an indication of whether all, most, some or just a few did so; I know some qualitative researchers do not like this, but it is becoming more common to see such qualifiers in qualitative medical/health papers).	As the reviewer recognises, there are differing views on presentation of qualitative findings. An indication of 'most' or 'some' or 'few' is used in the different Findings sub-sections. Where previously 'many' was indicated, this has been replaced with 'most' if appropriate to those particular findings.
d) In the discussion and abstract the issues of harm (i.e. people being worried about diabetes risk and what it means) and the lack of care pathways needs to be highlighted more.	Within the constraints of word limit, we have further highlighted the issues of potential harm and need for more appropriate care pathways in three places – in the Discussion (under Principal Findings, and under Implications for Practice) and also in the conclusion of the Abstract
7. As I added a few comments to the text, I add this. But these are just a few. a) Page 6: Why is this important? b) Page 9: How? More detail. c) Page 12: Why upper case? d) Page 23: Do you mean the quality of? Or the fidelity of delivery? That forms of delivery and contents are highly variable is a necessity, in my view?	a) '40% aged 30-59' is simply to provide further contextual detail on the whole locality population b) Details of sampling and interviewing are provided earlier in the 'sampling and data generation' section of Methods c) 'Attended' describing participants in quotation illustration Boxes could be either 'Attended' or 'attended' - depending on preferred style. d) Thank you – we have amended to clarify we do indeed mean the delivery and content of interventions are necessarily highly variable.